# A Study of Posterior Stability in Time-Series Latent Diffusion

**Yangming Li**[1]**, Yixin Cheng**[1,2,3]**, Mihaela van der Schaar**[1]
[1]DAMTP, University of Cambridge
[2]LIONS, EPFL
[3]Hithink Research
yl874@cam.ac.uk

## Abstract

Latent diffusion has achieved remarkable success in image generation, with high sampling efficiency. However, this framework might suffer from *posterior collapse* when applied to time series. In this work, we first show that latent diffusion with a collapsed posterior degenerates into a much weaker generative model: variational autoencoder (VAE). This finding highlights the significance of addressing the problem. We then introduce a principled method: *dependency measures*, which quantify the sensitivity of a recurrent decoder to input variables. Through this method, we confirm that posterior collapse seriously affects latent time-series diffusion on real time series. For example, the latent variable has an exponentially decreasing impact on the decoder over time. Building on our theoretical and empirical studies, we finally introduce a new framework: posterior-stable latent diffusion, which interprets the diffusion process as a type of variational inference. In this way, it eliminates the use of risky KL regularization and penalizes decoder insensitivity. Extensive experiments on multiple real time-series datasets show that our new framework is with a highly stable posterior and notably outperforms previous baselines in time series synthesis.

## 1 Introduction

Latent diffusion (Rombach et al., 2022) has shown strong performance in image synthesis (Podell et al., 2024), offering substantially faster sampling than standard diffusion models (Ho et al., 2020). However, when applied to time-series data, this framework might suffer from *posterior collapse* (Bowman et al., 2016), a common and important problem that happens to some generative models (e.g., autoencoder (Baldi, 2012; Lucas et al., 2019)), where the latent variable only captures limited information from the data. In that case, the decoder tends to ignore the variable during conditional generation. In this paper, we present a systematic analysis of posterior collapse in time-series latent diffusion and propose an improved framework that builds upon our analysis.

**Impact analysis of posterior collapse.** We first show that a strictly collapsed posterior reduces the latent diffusion to a vanilla variational autoencoder (VAE) (Kingma & Welling, 2013) in formulation, indicating that this problem renders the framework *inexpressive*, even weaker than a vanilla diffusion model. We then introduce a principled method: *dependency measure*, which quantifies the dependencies of an autoregressive decoder on the latent variable and the observed time series. Through performing empirical estimation of these measures, we confirm the latent variable has a nearly exponentially vanishing impact on the recurrent decoder, indicating that time-series latent diffusion indeed suffers from posterior collapse.

From this empirical study, we also observe an interesting symptom of posterior collapse: *dependency illusion*: when time series are randomly shuffled and thus lack structural dependencies, the estimated dependency measures show that the autoregressive decoder still heavily relies on previous observations (instead of the latent variable) for predicting the next one.

**Our posterior-stable latent diffusion.** We identify two main causes of posterior collapse in latent diffusion: KL regularization and strong decoder (Bowman et al., 2016). The first cause stems from the design of VAE, ensuring that the latent variable follows a simple prior distribution. Such regularization is in fact unnecessary in latent diffusion, as its diffusion component supports sampling latent variables from a complex distribution. The second cause is the absence of a mechanism that enforces decoder sensitivity to the latent variable.

In light of this analysis, we first propose to reinterpret the diffusion process as a form of variational inference, thereby eliminating the use for risky KL regularization in latent diffusion and allowing unrestricted prior distributions for latent variable sampling. To ensure that the decoder is sensitive to the latent variable, we also apply the diffusion process to simulate a collapsed posterior, imposing a significant penalty on dependency illusion.

**Contributions and roadmap.** In summary, our contributions are as follows:

- We provide a rigorous analysis of *posterior collapse* in latent diffusion, showing that a collapsed posterior will render it no more expressive than a simple VAE. We also introduce *dependency measures* that confirm the problem on real time-series data;

- We present a new type of time-series generative model: posterior-stable latent diffusion, which is free from risky KL regularization and ensures that the autoregressive decoder is sensitive to the input latent variable;

- We conduct extensive experiments on multiple real time-series datasets. The results demonstrate that our framework remains robust against posterior collapse and significantly outperforms a number of previous baselines.

The remainder of this paper is organized as follows. In Sec. 2, we review latent diffusion. Sec. 3 presents a systematic analysis of *posterior collapse* in time-series latent diffusion. Sec. 4 introduces our proposed framework to address this problem. Finally, Sec. 6 reports experimental results demonstrating the effectiveness of the framework.

## 2 BACKGROUND: LATENT DIFFUSION

The architecture of latent diffusion consists of two parts: 1) an autoencoder (Baldi, 2012) that maps high-dimensional or structured data into low-dimensional latent variables; 2) a diffusion model (Sohl-Dickstein et al., 2015) that learns the distribution of latent variables.

**Autoencoder.** The autoencoder is typically implemented as VAE (Kingma & Welling, 2013). Let $\mathbf{X}$ denote a free-form raw sample, following an underlying distribution $q^{\text{raw}}(\mathbf{X})$. The encoder $\mathbf{f}^{\text{enc}}$ of VAE aims to convert the sample into a low-dimensional vector $\mathbf{v} = \mathbf{f}^{\text{enc}}(\mathbf{X})$. Through a reparameterization trick, the vector can be used to compute latent variable $\mathbf{z}$ as

$$\mathbf{z} = \boldsymbol{\mu} + \text{diag}(\boldsymbol{\sigma}) \cdot \boldsymbol{\epsilon}, \boldsymbol{\epsilon} \sim \mathcal{N}(\mathbf{0}, \mathbf{I}), \quad \boldsymbol{\mu} = \mathbf{W}_\mu \mathbf{v}, \boldsymbol{\sigma} = \exp(\mathbf{W}_\sigma \mathbf{v}), \tag{1}$$

where $\mathbf{W}_\mu, \mathbf{W}_\sigma$ are learnable matrices, and $\text{diag}(\cdot)$ is an operation that casts a vector into a diagonal matrix. The above procedure, which differentially samples a latent variable $\mathbf{z}$ from the posterior $q^{\text{VI}}(\mathbf{z} \mid \mathbf{X}) = \mathcal{N}(\mathbf{z}; \boldsymbol{\mu}, \text{diag}(\boldsymbol{\sigma}^2))$, is termed *variational inference* (Blei et al., 2017). In VAE, the decoder $\mathbf{f}^{\text{dec}}$ is used to parameterize a conditional generation distribution $p^{\text{gen}}(\mathbf{X} \mid \mathbf{z})$, recovering the real sample $\mathbf{X}$ from latent variable $\mathbf{z}$.

The exact negative log-likelihood loss function of VAE is computationally infeasible, so its optimization relies on an upper bound of the loss:

$$\mathcal{L}^{\text{VAE}} = \mathbb{E}_{\mathbf{z} \sim q^{\text{VI}}(\mathbf{z} \mid \mathbf{X})}[-\ln p^{\text{gen}}(\mathbf{X} \mid \mathbf{z})] + \mathrm{D}_{\text{KL}}(q^{\text{VI}}(\mathbf{z} \mid \mathbf{X}) \,\|\, p^{\text{prior}}(\mathbf{z})), \tag{2}$$

where the prior distribution $p^{\text{prior}}(\mathbf{z})$ is typically set as a standard Gaussian $\mathcal{N}(\mathbf{0}, \mathbf{I})$. The last KL divergence term is to ensure that the prior $p^{\text{prior}}(\mathbf{z})$ is compatible with decoder $\mathbf{f}^{\text{dec}}$ at test time, though it is one cause of *posterior collapse* (Bowman et al., 2016).

**Diffusion model.** The diffusion model can be implemented as DDPM (Ho et al., 2020). The model consists of two Markov chains of $L \in \mathbb{N}^+$ steps. One of them is the diffusion process, which incrementally applies the forward transition kernel:

$$q^{\text{forw}}(\mathbf{z}^i \mid \mathbf{z}^{i-1}) = \mathcal{N}(\mathbf{z}^i; \sqrt{1 - \beta^i}\mathbf{z}^i, \beta^i\mathbf{I}), \tag{3}$$

to the latent variable $\mathbf{z}^0 := \mathbf{z} \sim q^{\text{latent}}(\mathbf{z})$, where $\beta^i, i \in [1, L]$ is some predefined variance schedule. Here the distribution of latent variable $q^{\text{latent}}(\mathbf{z})$ is defined as $\int q^{\text{VI}}(\mathbf{z} \mid \mathbf{X})q^{\text{raw}}(\mathbf{X})d\mathbf{X}$. The outcomes of this process form a series of new latent variables $\{\mathbf{z}^1, \mathbf{z}^2, \cdots, \mathbf{z}^L\}$, with the last one $\mathbf{z}^L$ approximately following a standard Gaussian $\mathcal{N}(\mathbf{0}, \mathbf{I})$ for $L \gg 1$.

The other is the reverse process, which iteratively applies the backward transition kernel:

$$p^{\text{back}}(\mathbf{z}^{i-1} \mid \mathbf{z}^i) = \mathcal{N}(\mathbf{z}^{i-1}; \boldsymbol{\mu}^{\text{back}}(\mathbf{z}^i, i), \sigma^i\mathbf{I}), \quad \boldsymbol{\mu}^{\text{back}}(\mathbf{z}^i, i) = \frac{1}{\sqrt{\alpha^i}}\left(\mathbf{z}^i - \beta^i\frac{\boldsymbol{\epsilon}^{\text{back}}(\mathbf{z}^i, i)}{\sqrt{1 - \bar{\alpha}^i}}\right), \tag{4}$$

where $\alpha^i = 1 - \beta^i$, $\bar{\alpha}^i = \prod_{k=1}^i \alpha^k$, $\boldsymbol{\epsilon}^{\text{back}}(\cdot)$ is a neural network, $\mathbf{z}^L$ is an initial sample drawn from $\sim \mathcal{N}(\mathbf{0}, \mathbf{I})$, and $\sigma^i$ is some backward variance schedule. The outcome of this process is a reversed sequence of latent variables $\{\mathbf{z}^{L-1}, \mathbf{z}^{L-2}, \cdots, \mathbf{z}^0\}$, where the last one $\mathbf{z}^0$ is expected to follow the distribution of real samples: $q^{\text{latent}}(\mathbf{z}^0)$.

To optimize the diffusion model, common practices adopt a loss function as below:

$$\mathcal{L}^{\text{DM}} = \mathbb{E}_{i, \mathbf{z}^0, \boldsymbol{\epsilon}}[\|\boldsymbol{\epsilon} - \boldsymbol{\epsilon}^{\text{back}}(\sqrt{\bar{\alpha}^i}\mathbf{z}^0 + \sqrt{1 - \bar{\alpha}^i}\boldsymbol{\epsilon}, i)\|^2], \tag{5}$$

where $\boldsymbol{\epsilon} \sim \mathcal{N}(\mathbf{0}, \mathbf{I})$, $\mathbf{z}_0 \sim q^{\text{latent}}(\mathbf{z}^0)$, $i \sim \mathcal{U}\{1, L\}$.

## 3 PROBLEM ANALYSIS

In this section, we first show the significant impact of *posterior collapse* on time-series latent diffusion. Then, we define proper measures that can empirically quantify this impact. Finally, we confirm that time-series diffusion indeed suffers from this problem on real datasets.

### 3.1 SIGNIFICANCE OF POSTERIOR COLLAPSE

Let us focus on time series $\mathbf{X} = [\mathbf{x}_1, \mathbf{x}_2, \cdots, \mathbf{x}_T]$, where every observation $\mathbf{x}_t, t \in [1, T]$ is a $D$-dimensional vector and $T$ denotes the number of observations. When applying latent diffusion to time series, a problem that might arise is *posterior collapse*, which occurs to many types of autoencoders, especially VAE. The problem can be formulated as follows.

**Problem formulation.** The posterior $q^{\text{VI}}(\mathbf{z} \mid \mathbf{X})$ of VAE is said to collapse if it reduces to the Gaussian prior $p^{\text{prior}}(\mathbf{z}) = \mathcal{N}(\mathbf{z}; \mathbf{0}, \mathbf{I})$, irrespective of the conditional $\mathbf{X}$:

$$q^{\text{VI}}(\mathbf{z} \mid \mathbf{X}) = p^{\text{prior}}(\mathbf{z}), \forall \mathbf{X} \in \mathbb{R}^{TD}.$$

In that case, the latent variable $\mathbf{z}$ contains no information about time series $\mathbf{X}$, otherwise the posterior distribution $q^{\text{VI}}(\mathbf{z} \mid \mathbf{X})$ would vary depending on different conditionals. Above is a strict definition. In practice, one is mostly faced with a situation where $q^{\text{VI}}(\mathbf{z} \mid \mathbf{X}) \approx p^{\text{prior}}(\mathbf{z})$ and it is still appropriate to say that the posterior collapses.

**Implications of posterior collapse.** A typical symptom of this problem is that, since the latent variable $\mathbf{z}$ carries very limited information of time series $\mathbf{X}$, the decoder $\mathbf{f}^{\text{dec}}$ tends to *ignore* this input variable $\mathbf{z}$, which is undesired for conditional generation $p^{\text{gen}}(\mathbf{X} \mid \mathbf{z})$. Besides this empirical finding from previous works, we find that posterior collapse also seriously impacts on the expressiveness of latent diffusion. Let us first see the below conclusion.

**Proposition 3.1** (Gaussian Latent Variables). *For standard latent diffusion, suppose its posterior $q^{\text{VI}}(\mathbf{z} \mid \mathbf{X})$ collapses, then the distribution $q^{\text{latent}}(\mathbf{z})$ of latent variable $\mathbf{z}$ will shape as a standard Gaussian $\mathcal{N}(\mathbf{0}, \mathbf{I})$, which is trivial for the diffusion model to approximate.*

Simply put, if posterior collapse happens, latent variable $\mathbf{z}$ will be Gaussian at test time. In that case, the diffusion model that aims to approximate the complex variable distribution $q^{\text{latent}}(\mathbf{z})$ is a redundant module. Therefore, posterior collapse reduces latent diffusion to a weak VAE, making it inexpressive. The proof for this conclusion is provided in Appendix A.

## 3.2 Introduction of Dependency Measures

A typical symptom (Bowman et al., 2016) of posterior collapse is that the input latent variable $\mathbf{z}$ loses control of decoder $\mathbf{f}^{\mathrm{dec}}$ during conditional generation $p^{\mathrm{gen}}(\mathbf{X} \mid \mathbf{z})$. To verify whether this is the case for time-series latent diffusion on real datasets, we introduce some proper measures that quantify the dependencies of decoder $\mathbf{f}^{\mathrm{dec}}$ on various inputs.

**Autoregressive decoder.** Consider that decoder $\mathbf{f}^{\mathrm{dec}}$ has an autoregressive structure, which conditions on latent variable $\mathbf{z}$ and prefix $\mathbf{X}_{1:t-1} = [\mathbf{x}_1, \mathbf{x}_2, \cdots, \mathbf{x}_{t-1}]$ to predict the next observation $\mathbf{x}_t$. With abuse of notation, we set $\mathbf{x}_0 = \mathbf{z}$ and formulate the decoder as

$$\mathbf{h}_t = \mathbf{f}^{\mathrm{dec}}(\mathbf{X}_{0:t-1}), \quad \mathbf{X}_{0:t-1} = [\mathbf{x}_0, \mathbf{x}_1, \mathbf{x}_2, \cdots, \mathbf{x}_{t-1}], \tag{6}$$

where the representation $\mathbf{h}_t, t \geq 1$ is linearly projected to multiple parameters (e.g., mean vector and covariance matrix) that determine the distribution $p^{\mathrm{gen}}(\mathbf{x}_t \mid \mathbf{z}, \mathbf{X}_{1:t-1})$ of some family (e.g., Gaussian). Examples of such a decoder include recurrent neural networks (RNN) (Hochreiter & Schmidhuber, 1997) and Transformer (Vaswani et al., 2017). We present the formulation details of these example in Appendix B.

**Dependency measure.** As mentioned, a clear symptom of posterior collapse is that the decoder $\mathbf{f}^{\mathrm{dec}}$ heavily relies on prefix $\mathbf{X}_{1:t-1}$ (especially the last observation $\mathbf{x}_{t-1}$) to compute the representation $\mathbf{h}_t$, ignoring the guidance of latent variable $\mathbf{x}_0 = \mathbf{z}$. The variable $\mathbf{z}$ that loses control of decoder $\mathbf{f}^{\mathrm{dec}}$ is undesired for conditional generation $p^{\mathrm{gen}}(\mathbf{X} \mid \mathbf{z})$.

Inspired by integrated gradients (Sundararajan et al., 2017), we present a technique: *dependency measure*, which quantifies the impacts of latent variable $\mathbf{x}_0 = \mathbf{z}$ and prefix $\mathbf{X}_{1:t-1}$ on decoder $\mathbf{f}^{\mathrm{dec}}$. Specifically, we first set a baseline input $\mathbf{O}_{0:t-1}$ as $[\mathbf{x}_0 = \mathbf{0}, \mathbf{x}_1 = \mathbf{0}, \cdots, \mathbf{x}_{t-1} = \mathbf{0}]$ and denote the term $\mathbf{f}^{\mathrm{dec}}(\mathbf{O}_{0:t-1})$ as $\widetilde{\mathbf{h}}_t$. Then, we parameterize a straight line $\boldsymbol{\gamma}(s) : [0, 1] \to \mathbb{R}^{tD}$ between the actual input $\mathbf{X}_{1:t-1}$ and the input baseline $\mathbf{O}_{0:t-1}$ as

$$\boldsymbol{\gamma}(s) = s\mathbf{X}_{0:t-1} + (1 - s)\mathbf{O}_{0:t-1} := [s\mathbf{x}_0, s\mathbf{x}_1, \cdots, s\mathbf{x}_{t-1}]. \tag{7}$$

Applying the chain rule in differential calculus, we have

$$\frac{d\mathbf{f}^{\mathrm{dec}}(\boldsymbol{\gamma}(s))}{ds} = \sum_{j=0}^{t-1} \sum_{k=1}^{k=D} \frac{d\mathbf{f}^{\mathrm{dec}}(\boldsymbol{\gamma}(s))}{d\gamma_{j,k}(s)} \frac{d\gamma_{j,k}(s)}{ds} = \sum_{j=0}^{t-1} \sum_{k=1}^{k=D} x_{j,k} \frac{d\mathbf{f}^{\mathrm{dec}}(\boldsymbol{\gamma}(s))}{d\gamma_{j,k}(s)}, \tag{8}$$

where $\gamma_{j,k}(s)$ denote the $k$-th dimension $s \cdot x_{j,k}$ of the $j$-th vector $s\mathbf{x}_j$ in point $\boldsymbol{\gamma}(s)$. With the above elements, we can define the below measures.

**Definition 3.2** (Dependency Measures). For an autoregressive decoder $\mathbf{f}^{\mathrm{dec}}$ that conditions on both latent variable $\mathbf{x}_0 = \mathbf{z}$ and the prefix $\mathbf{X}_{1:t-1}$ to compute representation $\mathbf{h}_t$, the dependency measure of every input variable $\mathbf{x}_j, j \in [0, t-1]$ to the decoder is defined as

$$m_{t,j} = \frac{1}{\|\mathbf{h}_t - \widetilde{\mathbf{h}}_t\|^2} < \mathbf{h}_t - \widetilde{\mathbf{h}}_t, \sum_k x_{j,k} \int_0^1 \frac{d\mathbf{f}^{\mathrm{dec}}(\boldsymbol{\gamma}(s))}{d\gamma_{j,k}(s)} ds >, \tag{9}$$

where $k \in [1, D]$ and operation $< \cdot, \cdot >$ represents the inner product. We name $m_{t,0}$ as the *global dependency* and $m_{t,t-j}, 1 \leq j < t$ as the *j-th order local dependency*.

We provide the derivation for dependency measure $m_{t,j}$ and detail its relations to integrated gradients in Appendix C. Plus, the integral term can be approximated as

$$\int_0^1 \frac{d\mathbf{f}^{\mathrm{dec}}(\boldsymbol{\gamma}(s))}{d\gamma_{j,k}(s)} ds \approx \frac{1}{|\mathcal{S}|} \sum_{s \in \mathcal{S}} \frac{d\mathbf{f}^{\mathrm{dec}}(\boldsymbol{\gamma}(s))}{d\gamma_{j,k}(s)}, \tag{10}$$

where $\mathcal{S}$ is the set of independent samples drawn from uniform distribution $\mathcal{U}\{0, 1\}$. According to the law of large numbers (Sedor, 2015), this approximation is unbiased and gets more accurate for a bigger sample set $|\mathcal{S}|$. Lastly, the defined measures have the following properties.

**Proposition 3.3** (Signed and Normalization Properties). *The dependency measure $m_{t,j}, \forall j \in [0, t-1]$ is a signed measure and always satisfies $\sum_{j=0}^{t-1} m_{t,j} = 1$.*

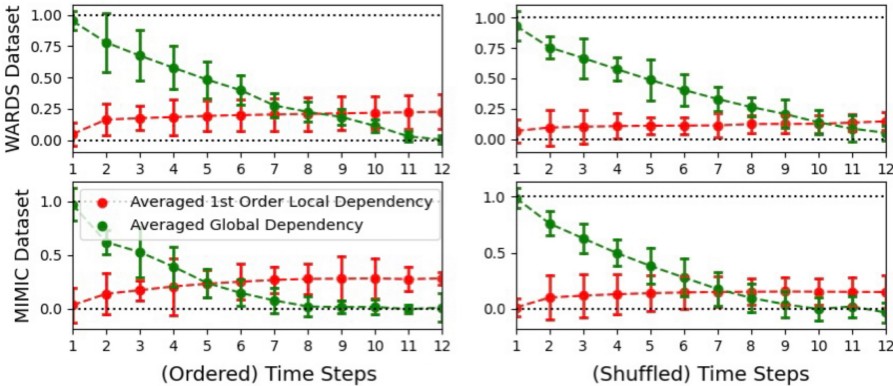

Figure 1: Dependency measures $m_{t,0}, m_{t,t-1}$ averaged over 500 multivariate time series, with 3 standard deviations as the error bars. We can see that the latent variable $\mathbf{z}$ of latent diffusion has a vanishing impact on the decoder $\mathbf{f}^{\text{dec}}$, a typical symptom of *posterior collapse*. We also observe a phenomenon of *dependency illusion* in the case of shuffled time series.

We can see that the measure $m_{t,j}$ can be either positive or negative, with a normalized sum over the subscript $j$ as 1. If $m_{t,j} \geq 0$, then we say that vector $\mathbf{x}_j$ has a positive impact on the decoder $\mathbf{f}^{\text{dec}}$: the bigger is $m_{t,j}$, the larger is such an impact; Similarly, if $m_{t,j} < 0$, then the vector $\mathbf{x}_j$ has a negative impact on the decoder: the smaller is $m_{t,j}$, the greater is the negative influence. It is also not hard to understand that there exists a negative impact. For example, the latent variable $\mathbf{z} \sim q^{\text{latent}}(\mathbf{z})$ might be an outlier for the decoder $\mathbf{f}^{\text{dec}}$, which locates at a low-density region in the prior distribution $q^{\text{prior}}(\mathbf{z})$. We provide the proof for this conclusion in Appendix D.

### 3.3 EMPIRICAL DEPENDENCY ESTIMATIONS

We are mainly interested in two types of defined measures. One is the *global dependency* $m_{t,0}$, which estimates the impact of latent variable $\mathbf{x}_0 = \mathbf{z}$ on the decoder $\mathbf{f}^{\text{dec}}$; The other is the *first-order local dependency* $m_{t,t-1}$, which estimates the dependency of decoder $\mathbf{f}^{\text{dec}}$ on the last observation $\mathbf{x}_{t,t-1}$. In this part, we empirically estimate these measures, with the aims to confirm that *posterior collapse* occurs and show its impacts.

**Experiment setup.** We adopt two real time-series datasets: WARDS (Alaa et al., 2017b) and MIMIC (Johnson et al., 2016). To study the case where time series have no structural dependencies, we also try randomly shuffling the time steps of ordered time series. We train latent diffusion models on those datasets and sample time series from the models.

**Insightful results.** Fig. 1 illustrates the estimated dependency measures $m_{t,0}, m_{t,t-1}$ of time-series latent diffusion averaged over 500 samples. We can see that, for both ordered and shuffled time series, the global dependency $m_{t,0}$ exponentially converges to 0 with increasing time step $t$, indicating posterior collapse: latent variable $\mathbf{z}$ loses control of the decoder $\mathbf{f}^{\text{dec}}$.

More interesting results are shown in the right two subfigures. When adjacent observations $\mathbf{x}_{t-1}, \mathbf{x}_t$ are not much correlated in shuffled time series, we still observe that the first-order measure $m_{t,t-1}$ is notably different from 0 (e.g., around 0.1 to 0.2). This phenomenon might arise as neural networks overfit and we term it as *dependency illusion: different observations $\mathbf{x}_s, \mathbf{x}_t, s < t$ in time series $\mathbf{X}$ are totally or almost independent, but the decoder $\mathbf{f}^{\text{dec}}$ still highly relies on $\mathbf{x}_s$ to predict $\mathbf{x}_t$ (i.e., high dependency measure $m_{t,s}$).*

## 4 METHOD: POSTERIOR-STABLE LATENT DIFFUSION

In this section, we first analyze the potential causes of posterior collapse in time-series latent diffusion. Based on the discussion, we then introduce a new framework, which extends from latent diffusion but is free from the problem.

## 4.1 Causes of Posterior Collapse in Latent Diffusion

Previous works (Semeniuta et al., 2017; Alemi et al., 2018) have identified two main causes of posterior collapse: *KL-divergence term* and *strong decoder*. For time-series latent diffusion, we will explain as below that those causes indeed exist, but are in fact avoidable.

**Unnecessary KL regularization.** The KL-divergence term $\mathrm{D}_{\mathrm{KL}}(q^{\mathrm{VI}}(\mathbf{z} \mid \mathbf{X}) \parallel p^{\mathrm{prior}}(\mathbf{z}))$ in Eq. (2) moves the posterior $q^{\mathrm{VI}}(\mathbf{z} \mid \mathbf{X})$ towards prior $p^{\mathrm{prior}}(\mathbf{z})$, which might incur posterior collapse by definition (see Sec. 3.1). In fact, this term is tailored for VAE, such that it is proper to sample latent variable $\mathbf{z}$ from a Gaussian prior $p^{\mathrm{prior}}(\mathbf{z})$ at test time. *However, the variable $\mathbf{z}$ in latent diffusion is sampled from the diffusion model, which can approximate a non-Gaussian prior distribution.* In this regard, we can see that KL regularization is not very necessary in latent diffusion, which also results in a limited prior $p^{\mathrm{prior}}(\mathbf{z})$.

**Time-series decoders are more vulnerable.** The strong decoder is also a cause of posterior collapse, which happens to sequence autoencoders (Bowman et al., 2016; Eikema & Aziz, 2019). Time series $\mathbf{X} \in \mathbb{R}^{TD}$ has a clear temporal structure, and thus its decoding is autoregressive, resulting in a strong decoder $\mathbf{f}^{\mathrm{dec}}$. In contrast, the decoder in image-based latent diffusion, is mainly configured as a feedforward neural network (Svozil et al., 1997), such as U-Net (Ronneberger et al., 2015). *This type of feedforward decoder is naturally sensitive to the input variables*, so the original design of latent diffusion did not consider address the possible insensitivity.

## 4.2 New Framework with a Stable Posterior

In light of the above discussion, we introduce a type of posterior-stable latent diffusion that eliminates the use of risky KL regularization, permit a free-form prior distribution $p^{\mathrm{prior}}(\mathbf{z})$, and increase the sensitivity of decoder $\mathbf{f}^{\mathrm{dec}}$ to latent variable $\mathbf{z}$.

Importantly, we notice a conclusion (Ho et al., 2020) for the diffusion process (i.e., Eq. (3)):

$$q^{\mathrm{forw}}(\mathbf{z}^i \mid \mathbf{z}^0) = \mathcal{N}(\mathbf{z}^i; \sqrt{\bar{\alpha}^i}\mathbf{z}^0, (1 - \bar{\alpha}^i)\mathbf{I}), \tag{11}$$

where the coefficient $\bar{\alpha}^i$ monotonically decreases from 1 to approximately 0 for $i \in [0, L]$. In this sense, suppose the initial variable $\mathbf{z}^0$ is set as $\mathbf{v} = \mathbf{f}^{\mathrm{enc}}(\mathbf{X})$, then we can infer that the random variable $\mathbf{z}^i \sim q^{\mathrm{forw}}(\mathbf{z}^i \mid \mathbf{z}^0)$ contains $\bar{\alpha}^i \times 100\%$ information about the vector $\mathbf{v}$, with $(1 - \bar{\alpha}^i) \times 100\%$ pure noise. For $i \rightarrow 0$, the diffusion process is similar to the *variational inference* (i.e., Eq. (1)) of VAE, adding slight Gaussian noise to the encoder output $\mathbf{v}$. For $i \rightarrow T$, the variable $\mathbf{z}^i$ simulates the problem of *posterior collapse* since $q^{\mathrm{forw}}(\mathbf{z}^i \mid \mathbf{z}^0) \approx \mathcal{N}(\mathbf{z}^i; \mathbf{0}, \mathbf{I})$.

**Diffusion process as variational inference.** Considering the above facts, we first treat the starting few iterations of the diffusion process as the *variational inference*. Specifically, with a fixed small integer $N \ll L$, we sample a number $i$ from uniform distribution $\mathcal{U}\{0, N\}$ and let the diffusion process convert the encoder output $\mathbf{v} = \mathbf{f}^{\mathrm{enc}}(\mathbf{X})$ into the latent variable:

$$\mathbf{z} = \mathbf{z}^i \sim q^{\mathrm{forw}}(\mathbf{z}^i \mid \mathbf{z}^0), \mathbf{z}^0 = \mathbf{v}. \tag{12}$$

In terms of the formerly defined generation distribution $p^{\mathrm{gen}}(\mathbf{X} \mid \mathbf{z})$ (parameterized by the decoder $\mathbf{f}^{\mathrm{dec}}$), a negative log-likelihood loss $\mathcal{L}^{\mathrm{VI}}$ is incurred as

$$\mathcal{L}^{\mathrm{VI}} = \mathbb{E}_{i \sim \mathcal{U}\{0,N\}, \mathbf{z}^0}[-\bar{\alpha}^{\gamma i} \ln p^{\mathrm{gen}}(\mathbf{X} \mid \mathbf{z} = \mathbf{z}^i)], \tag{13}$$

where $\gamma \in \mathbb{N}^+, \gamma N \leq L$ is a hyper-parameter, with the aim to reduce the impact of a very noisy latent variable $\mathbf{z}$. As multiplier $\gamma$ increases, the weight $\bar{\alpha}^{\gamma i}$ decreases.

Similar to VAE, the variational inference in our framework also leads the latent variable $\mathbf{z}$ to be *smooth* (Bowman et al., 2016) in its effect on decoder $\mathbf{f}^{\mathrm{dec}}$. However, our framework is free from the KL-divergence term $\mathrm{D}_{\mathrm{KL}}(q^{\mathrm{VI}}(\mathbf{z} \mid \mathbf{X}) \parallel p^{\mathrm{prior}}(\mathbf{z}))$ of VAE (i.e., one cause of the *posterior collapse*), since we can facilitate $\mathbf{z} \sim q^{\mathrm{latent}}(\mathbf{z})$ at test time through applying the reverse process of the diffusion model (i.e., Eq. (4)) to sample variable $\mathbf{z}^i, i \in [0, N]$.

| **Algorithm 1** Training | **Algorithm 2** Sampling |
|---|---|
| 1: **repeat** | 1: $\mathbf{z}_L \sim p^{\text{back}}(\mathbf{z}_L) = \mathcal{N}(\mathbf{0}, \mathbf{I})$ |
| 2: Encoding time-series sample: $\mathbf{v} = \mathbf{f}^{\text{enc}}(\mathbf{X})$ | 2: Set stop time: $i \sim \mathcal{U}\{0, N\}$ |
| 3: $\mathbf{z}^j \sim q^{\text{forw}}(\mathbf{z}^j \mid \mathbf{z}^0 = \mathbf{v}), j \sim \mathcal{U}\{0, N\}$ | 3: **for** $l = L, L-1, \dots, i+1$ **do** |
| 4: $\widehat{\mathcal{L}}^{\text{VI}} = -\bar{\alpha}^{\gamma j} \ln p^{\text{gen}}(\mathbf{X} \mid \mathbf{z} = \mathbf{z}^j)$ | 4: $\mathbf{z}^{l-1} \sim p^{\text{back}}(\mathbf{z}^{l-1} \mid \mathbf{z}^l)$ |
| 5: $i \sim \mathcal{U}\{j, L\}, \boldsymbol{\epsilon} \sim \mathcal{N}(\mathbf{0}, \mathbf{I})$ | 5: **end for** |
| 6: $\widehat{\mathcal{L}}^{\text{DM}} = \|\boldsymbol{\epsilon} - \boldsymbol{\epsilon}^{\text{back}}(\sqrt{\bar{\alpha}^i}\mathbf{z}^j + \sqrt{\cdot}\boldsymbol{\epsilon}, i)\|^2$ | 6: Conditional generation: $p^{\text{gen}}(\widehat{\mathbf{X}} \mid \mathbf{z} = \mathbf{z}^i)$ |
| 7: $\mathbf{z}^k \sim q^{\text{forw}}(\mathbf{z}^k \mid \mathbf{z}^0 = \mathbf{v}), k \sim \mathcal{U}\{M, L\}$ | 7: **return** Time series $\widehat{\mathbf{X}}$ |
| 8: $\widehat{\mathcal{L}}^{\text{CS}} = (1 - \bar{\alpha}^{\lceil \frac{k}{\eta} \rceil}) \ln p^{\text{gen}}(\mathbf{X} \mid \mathbf{z} = \mathbf{z}^k)$ | |
| 9: Gradient descent with $\nabla(\widehat{\mathcal{L}}^{\text{VI}} + \widehat{\mathcal{L}}^{\text{DM}} + \widehat{\mathcal{L}}^{\text{CS}})$ | |
| 10: **until** converged | |

**Diffusion process for collapse simulation.** Then, we apply the last few iterations of the diffusion process to simulate *posterior collapse*, with the aims of increasing the impact of latent variable $\mathbf{z}$ on conditional generation $p^{\text{gen}}(\mathbf{X} \mid \mathbf{z})$ and reducing *dependency illusion*.

Following our previous *variational inference*, we set $\mathbf{z}^0 = \mathbf{f}^{\text{enc}}(\mathbf{X})$ and apply the diffusion process to cast the initial variable $\mathbf{z}^0$ into a highly noisy variable $\mathbf{z}^i, i \to L$. Considering that the variable $\mathbf{z}^i$ contains little information about the encoder output $\mathbf{f}^{\text{enc}}(\mathbf{X})$, it is unlikely that the decoder $\mathbf{f}^{\text{dec}}$ can recover time series $\mathbf{X}$ from variable $\mathbf{z}^i$, otherwise there is *posterior collapse* or *dependency illusion*. In this sense, we have the following regularization:

$$\mathcal{L}^{\text{CS}} = \mathbb{E}_{i, \mathbf{z}^i}[(1 - \bar{\alpha}^{\lceil \frac{i}{\eta} \rceil}) \ln p^{\text{gen}}(\mathbf{X} \mid \mathbf{z} = \mathbf{z}^i)], \tag{14}$$

which penalizes the model for having a high conditional density $p^{\text{gen}}(\mathbf{X} \mid \mathbf{z})$ for non-informative latent variable $\mathbf{z} = \mathbf{z}^i, i \in [M, L]$. Here $M \in \mathbb{N}^+$ is close to $L$, $i \sim \mathcal{U}\{M, L\}$, $\lceil \cdot \rceil$ is the ceiling function, and $\eta \geq 1$ is set to reduce the impact of informative variable $\mathbf{z}^i$.

For a *strong decoder* $\mathbf{f}^{\text{dec}}$, the regularization $\mathcal{L}^{\text{CS}}$ will impose a heavy penalty if the decoder solely relies on previous observations $\{\mathbf{x}_k \mid k < j\}$ to predict an observation $\mathbf{x}_j$. In that situation, even if the latent variable $\mathbf{z}$ contains very limited information about the raw data $\mathbf{X}$, a high prediction probability will still be assigned to the observation $\mathbf{x}_j$.

**Training, inference, and running times.** While our new framework extends from latent diffusion, its training and inference procedures are very different. We respectively depict the two procedures in Algorithm 1 and Algorithm 2. For training, the key points are to compute three loss functions: $\widehat{\mathcal{L}}^{\text{VI}}$ for likelihood maximization, $\widehat{\mathcal{L}}^{\text{DM}}$ for training diffusion models, and $\widehat{\mathcal{L}}^{\text{CS}}$ for collapse regularization. For inference, the main difference is that the stopping time of the backward process is not 0, but a random variable. From these pseudo codes, we can see that our framework is almost as efficient as vanilla latent diffusion. We provide an in-depth analysis and empirical experiments about the running times of our framework in Appendix F.2.

## 5 RELATED WORK

Besides latent diffusion, our paper is related to previous methods that aim to mitigate the problem of *posterior collapse* for VAE (Kingma & Welling, 2013). In the following, we will first briefly introduce those baselines, explaining their limitations, and then discuss some other types of time-series generative models.

**Existing methods for mitigating posterior collapse.** KL annealing (Bowman et al., 2016; Fu et al., 2019; Ichikawa & Hukushima, 2024) is to assign an adaptive weight to control the effect of KL-divergence term, so that VAE is unlikely to fall into the local optimum of posterior collapse at the initial optimization stage. With a similar idea, there are other types of regularization (e.g., mutual information constraints (Melis et al., 2022)) in the literature. Such methods can only mitigate the problem to some degree, but cannot fully reduce it.

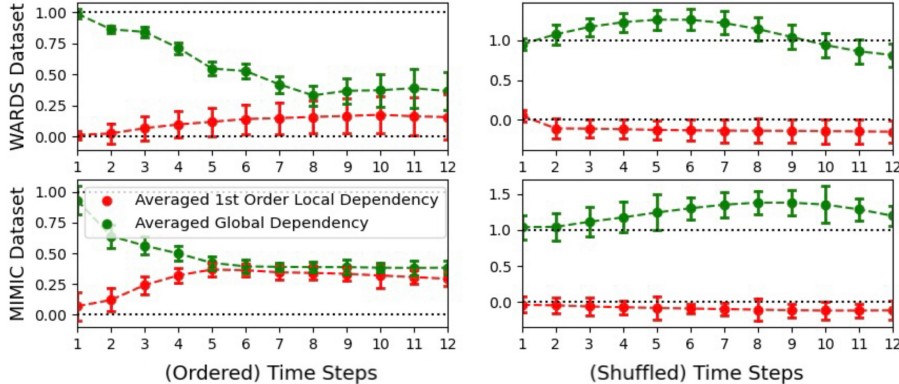

Figure 2: The results of averaged dependency measures and error bars for our framework, which should be compared with those (e.g., Fig. 1) of latent diffusion, showing that our framework has a *stable posterior* and is without *dependency illusion*.

Decoder weakening (Lu et al., 2021; Kinoshita et al., 2023) is also a popular class of methods to address posterior collapse. A well-known example is Variable Masking (Semeniuta et al., 2017), which randomly masks input observations to the autoregressive decoder, such that the decoder is forced to rely more on the latent variable for predicting the next observation. However, this method will make the model inexpressive since the decoder is weakened.

To improve the impact of latent variables on the recurrent decoder, a method called skip connections (Dieng et al., 2019; Fu et al., 2024) directly feeds the latent variable into the decoder at every step, not only at the first step. However, the latent variable in that case acts as a constant input signal at every time step, so the decoder will still tend to ignore this redundant information.

**Other time-series generative models.** In addition to latent diffusion, prior studies have proposed various alternative models for time-series generation. For example, Neural ODE Rubanova et al. (2019); Li et al. (2024) that works well on irregular time series, and TimeGAN (Yoon et al., 2019) that is based on Generative Adversarial Networks (GAN) (Goodfellow et al., 2020). We have included these three generative baselines in Table 2 for comparison.

## 6 EXPERIMENTS

### 6.1 STABLE POSTERIOR OF OUR FRAMEWORK

To show that our framework has a stable posterior $q^{\mathrm{VI}}(\mathbf{z} \mid \mathbf{X})$, we follow the same experiment setup (e.g., datasets) as Sec. 3.3 and average the dependency measures $m_{t,0}, m_{t,t-1}$ over 500 sampled time series. The results are illustrated in Fig. 2. For ordered time series in the left two subfigures, we can see that, while the global dependency $m_{t,0}$ still decreases with increasing time step $t$, it converges into a value around 0.5, which is also a bit higher than the converged first-order local dependency $m_{t,t-1}$. These results indicate that latent variable $\mathbf{z}$ in our framework maintains its control of decoder $\mathbf{f}^{\mathrm{dec}}$ during the entire conditional generation process $p^{\mathrm{gen}}(\mathbf{X} \mid \mathbf{z})$.

For shuffled time series in the right two subfigures, we can see that the global dependency $m_{t,0}$ is always around or above 1, and the local dependency $m_{t,t-1}$ is negative most of the time. These results indicate that the decoder $\mathbf{f}^{\mathrm{dec}}$ only relies on latent variable $\mathbf{z}$ and the context $\mathbf{x}_{t-1}$ even has a negative impact on conditional generation $p^{\mathrm{gen}}(\mathbf{X} \mid \mathbf{z})$, suggesting our framework is without *dependency illusion*. Based on all our findings, we conclude that: compared with latent diffusion (Fig. 1), our framework is free from posterior collapse.

### 6.2 PERFORMANCES IN TIME SERIES GENERATION

In this part, we aim to verify that our framework improves latent diffusion in terms of time series generation, which is promising since it is free from *posterior collapse*. We also include some other

| Model | Backbone | MIMIC | WARDS | Earthquakes |
|---|---|---|---|---|
| Latent Diffusion | LSTM | 5.19 | 7.52 | 5.87 |
| Latent Diffusion w/ KL Annealing | LSTM | 4.28 | 5.74 | 3.88 |
| Latent Diffusion w/ Variable Masking | LSTM | 4.73 | 6.01 | 4.26 |
| Latent Diffusion w/ Skip Connections | LSTM | 3.91 | 4.95 | 3.74 |
| Our Framework | LSTM | **2.29** | **3.16** | **2.67** |
| Latent Diffusion | Transformer | 5.02 | 7.46 | 5.91 |
| Latent Diffusion w/ KL Annealing | Transformer | 4.31 | 5.54 | 3.51 |
| Latent Diffusion w/ Variable Masking | Transformer | 4.42 | 5.97 | 4.45 |
| Latent Diffusion w/ Skip Connections | Transformer | 3.75 | 4.67 | 3.69 |
| Our Framework | Transformer | **2.13** | **3.01** | **2.49** |

Table 1: Wasserstein distances of different models on real time-series datasets. The lower the distance metric, the better the generation quality. *More results from other datasets, with another evaluation metric, are placed in Table 5 of Appendix F.3.*

| Model | MIMIC | Earthquakes |
|---|---|---|
| Latent Diffusion (Rombach et al., 2022) | 5.02 | 5.91 |
| Latent Diffusion w/ Mutual Information Constraints (Melis et al., 2022) | 3.59 | 3.85 |
| Latent Diffusion w/ Inverse Lipschitz Constraint (Kinoshita et al., 2023) | 3.01 | 3.42 |
| Neural STPP (Chen et al., 2021) | 5.13 | 5.82 |
| Neural Latent Dynamic (Li et al., 2024) | 4.31 | 5.12 |
| Frequency Diffusion (Crabbé et al., 2024) | 4.56 | 5.07 |
| Our Framework | **2.13** | **2.49** |

Table 2: Comparison with more baselines, including recent new methods to mitigate *posterior collapse* and other types of time-series generative models. Both latent diffusion baselines and our model are with Transformer as the backbone.

previous methods that can mitigate *posterior collapse*, including KL annealing (Fu et al., 2019), variable masking (Bowman et al., 2016), and skip connections (Dieng et al., 2019). We adopt the Wasserstein distances (Bischoff et al., 2024) as the metric.

The experiment results on three commonly used time-series datasets are shown in Table 1. From the results, we can see that, regardless of the used dataset and the backbone of autoencoder, our framework significantly outperforms latent diffusion and the baselines, which strongly confirms our intuition. For example, with the backbone of Transformer, our framework achieves 2.53 points lower than latent diffusion w/ KL Annealing on the WARDS dataset.

We have also compared our model with more recent baselines, including non-latent diffusion generative models and new methods to mitigate posterior collapse. The results are shown in Table 2, which further confirm the effectiveness of our framework.

### 6.3 More Results in the Appendix

Due to space limitations in the main text, additional experimental results are provided in the appendix: more time-series datasets and an alternative evaluation metric (Appendix F.3), results on text and image modalities (Appendix F.4), ablation studies (Appendix F.1), and running-time analysis (Appendix F.2). Details of the experimental setup are also included in Appendix E.

## 7 Conclusion

In this paper, we first provide a solid analysis of *posterior collapse* in latent diffusion, showing that the problem will reduce it to a weak generative model (i.e., VAE). Then, we introduce a quantitative method: *dependency measures*, confirming that time-series latent diffusion indeed has an unstable posterior on real data. Lastly, we introduce a new framework that eliminates the use of risky KL regularization, permits an unlimited prior distribution, and ensures that the decoder is sensitive to the input latent variable. Extensive experiments on multiple real time-series datasets show that our framework has a stable posterior and outperforms previous baselines.

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

# A   THE IMPACT OF POSTERIOR COLLAPSE

Under the assumption of *posterior collapse*, the below equality:

$$q^{\mathrm{VI}}(\mathbf{z} \mid \mathbf{X}) = p^{\mathrm{prior}}(\mathbf{z}) = \mathcal{N}(\mathbf{z}; \mathbf{0}, \mathbf{I}), \tag{15}$$

holds for any latent variable $\mathbf{z} \in \mathbb{R}^D$ and any conditional $\mathbf{X} \in \mathbb{R}^{TD}$. Then, note that

$$
\begin{aligned}
q^{\mathrm{latent}}(\mathbf{z}) &= \int q^{\mathrm{VI}}(\mathbf{z} \mid \mathbf{X}) q^{\mathrm{raw}}(\mathbf{X}) d\mathbf{X} = \int \mathcal{N}(\mathbf{z}; \mathbf{0}, \mathbf{I}) q^{\mathrm{raw}}(\mathbf{X}) d\mathbf{X} \\
&= \mathcal{N}(\mathbf{z}; \mathbf{0}, \mathbf{I}) \int q^{\mathrm{raw}}(\mathbf{X}) d\mathbf{X} = \mathcal{N}(\mathbf{z}; \mathbf{0}, \mathbf{I}),
\end{aligned}
\tag{16}
$$

which is exactly our claim.

# B   RECURRENT ENCODERS

We mainly implement the backbone of decoder $\mathbf{f}^{\mathrm{dec}}$ as LSTM (Hochreiter & Schmidhuber, 1997) or Transformer (Vaswani et al., 2017). In the former case, we apply the latent variable $\mathbf{z}$ to initialize LSTM and condition it on prefix $\mathbf{X}_{1:t-1}$ to compute the representation $\mathbf{h}_t$. Formally, the LSTM-based decoder $\mathbf{f}^{\mathrm{dec}}$ is as

$$
\begin{cases}
\mathbf{s}_t = \mathrm{LSTM}(\mathbf{s}_{t-1}, \mathbf{x}_{t-1}), \forall t \geq 1 \\
\mathbf{h}_t = \mathbf{W}_f^2 \tanh(\mathbf{W}_f^1 \mathbf{s}_t)
\end{cases}, \tag{17}
$$

where $\mathbf{s}_t$ is the state vector of LSTM and $\mathbf{W}_f^2, \mathbf{W}_f^1$ are learnable matrices. In particular, for the corner case $t = 1$, we fix $\mathbf{s}_0, \mathbf{x}_0$ as zero vectors.

In the later case, we just treat latent variable $\mathbf{z}$ as $\mathbf{x}_0$. Therefore, we have

$$
\begin{cases}
[\mathbf{s}_{t-1}, \mathbf{s}_{s-2}, \cdots, \mathbf{s}_0] = \mathrm{Transformer}(\mathbf{x}_{t-1}, \mathbf{x}_{t-2}, \cdots, \mathbf{x}_0) \\
\mathbf{h}_t = \mathbf{W}_f^2 \tanh(\mathbf{W}_f^1 \mathbf{s}_{t-1})
\end{cases}, \tag{18}
$$

where the subscript alignment results from self-attention mechanism.

# C   DERIVATION OF DEPENDENCY MEASURES

Integrated gradient (Sundararajan et al., 2017) is a very effective method of feature attributions. Our dependency measures can be regarded as its extension to the case of sequence data and vector-valued neural networks. In the following, we provide the derivation of dependency measures.

For the computation $\mathbf{h}_t = \mathbf{f}^{\mathrm{dec}}(\mathbf{X}_{0:t-1})$, suppose the output of decoder $\mathbf{f}^{\mathrm{dec}}$ at origin $\mathbf{O}_{0:t-1}$ is $\widetilde{\mathbf{h}}_t$, then we apply the fundamental theorem of calculus as

$$\mathbf{h}_t - \widetilde{\mathbf{h}}_t = \int_0^1 \frac{d\mathbf{f}^{\mathrm{dec}}(\boldsymbol{\gamma}(s))}{ds} ds, \tag{19}$$

where $\boldsymbol{\gamma}(s)$ is a straight line connecting the origin $\mathbf{O}_{0:t-1}$ and the input $\mathbf{X}_{0:t-1}$ as $\boldsymbol{\gamma}(s) = s\mathbf{X}_{0:t-1} + (1-s)\mathbf{O}_{0:t-1}$. Based on the chain rule, the above equality can be expanded as

$$
\begin{aligned}
\mathbf{h}_t - \widetilde{\mathbf{h}}_t &= \int_0^1 \sum_{j=0}^{t-1} \sum_{k=1}^{k=D} \frac{d\mathbf{f}^{\mathrm{dec}}(\boldsymbol{\gamma}(s))}{d\gamma_{j,k}(s)} \frac{d\gamma_{j,k}(s)}{ds} ds \\
&= \sum_{j=0}^{t-1} \left( \int_0^1 \sum_{k=1}^{k=D} x_{j,k} \frac{d\mathbf{f}^{\mathrm{dec}}(\boldsymbol{\gamma}(s))}{d\gamma_{j,k}(s)} ds \right),
\end{aligned}
\tag{20}
$$

where $\gamma_{j,k}(s)$ denote the $k$-th dimension $s \cdot x_{j,k}$ of the $j$-th vector $s\mathbf{x}_j$ in point $\boldsymbol{\gamma}(s)$. Intuitively speaking, every term inside the outer sum operation $\sum_{j=0}^{t-1}$ represents the additive contribution of variable $\mathbf{x}_j$ (to the output difference $\mathbf{h}_t - \widetilde{\mathbf{h}}_t$) along the integral line $\boldsymbol{\gamma}(s)$.

| Model | Backbone | $N$ for $\mathcal{L}^{\text{VI}}$ | $M$ for $\mathcal{L}^{\text{CS}}$ | Diffusion Iterations $L$ | MIMIC | WARDS |
|---|---|---|---|---|---|---|
| Latent Diffusion | Transformer | − | − | 1000 | 5.02 | 7.46 |
| LD w/ Skip Connections | Transformer | − | − | 1000 | 3.75 | 4.67 |
| Our Framework | Transformer | 50 | 100 | 1000 | **2.13** | **3.01** |
| Our Framework | Transformer | 50 | 50 | 1000 | 2.59 | 3.32 |
| Our Framework | Transformer | 50 | 150 | 1000 | 2.71 | 3.46 |
| Our Framework | Transformer | 50 | 200 | 1000 | 2.83 | 3.75 |
| Our Framework | Transformer | 10 | 100 | 1000 | 2.31 | 3.16 |
| Our Framework | Transformer | 100 | 100 | 1000 | 2.38 | 3.24 |
| Our Framework | Transformer | 150 | 100 | 1000 | 2.75 | 3.41 |

Table 3: Ablation studies of the hyper-parameters $N, M$, which are respectively used in the estimations of likelihood loss $\mathcal{L}^{\text{VI}}$ and collapse penalty $\mathcal{L}^{\text{CS}}$. Here LD is short for latent diffusion and the symbol $-$ means "Not Applicable".

To simplify the notation, we denote the mentioned term as

$$\mathbf{m}_{t,j} = \int_0^1 \sum_{k=1}^{k=D} x_{j,k} \frac{d\mathbf{f}^{\text{dec}}(\boldsymbol{\gamma}(s))}{d\gamma_{j,k}(s)} ds. \tag{21}$$

Since $\mathbf{m}_{t,j}$ is a vector, we map the new term to a scalar and re-scale it as

$$m_{t,j} = \frac{< \mathbf{m}_{t,j}, \mathbf{h}_t - \widetilde{\mathbf{h}}_t >}{< \mathbf{h}_t - \widetilde{\mathbf{h}}_t, \mathbf{h}_t - \widetilde{\mathbf{h}}_t >}, \tag{22}$$

which is exactly our definition of the dependency measure.

## D  PROPERTIES OF OF DEPENDENCY MEASURES

Firstly, in terms of Eq. (22), it is obvious that the dependency measure $m_{t,j}$ is signed: the measure can be either positive or negative. Then, based on Eq. (20), we have

$$\mathbf{h}_t - \widetilde{\mathbf{h}}_t = \sum_{j=0}^{t-1} \mathbf{m}_{t,j}. \tag{23}$$

By taking an inner product with the vector $\mathbf{h}_t - \widetilde{\mathbf{h}}_t$ at both sides, we get

$$< \mathbf{h}_t - \widetilde{\mathbf{h}}_t, \mathbf{h}_t - \widetilde{\mathbf{h}}_t > = < \sum_{j=0}^{t-1} \mathbf{m}_{t,j}, \mathbf{h}_t - \widetilde{\mathbf{h}}_t > = \sum_{j=0}^{t-1} < \mathbf{m}_{t,j}, \mathbf{h}_t - \widetilde{\mathbf{h}}_t > . \tag{24}$$

By rearranging the term, we finally arrive at

$$1 = \sum_{j=0}^{t-1} \frac{< \mathbf{m}_{t,j}, \mathbf{h}_t - \widetilde{\mathbf{h}}_t >}{< \mathbf{h}_t - \widetilde{\mathbf{h}}_t, \mathbf{h}_t - \widetilde{\mathbf{h}}_t >} = \sum_{j=0}^{t-1} m_{t,j}, \tag{25}$$

which is exactly our claim.

## E  EXPERIMENT DETAILS

We have adopted three widely used real-world time-series datasets for both analysis and model evaluation, including MIMIC (Johnson et al., 2016), WARDS (Alaa et al., 2017a), and Earthquakes (U.S. Geological Survey, 2020). For the first two datasets, we extract the observations of the first 12 hours, with the top 5 features that have the highest variances forming multivariate time series. For MIMIC, we specially simplify it into a version of univariate time series for the illustration purpose, which is only used in the experiments shown in Fig. 3. All other experiments are about multivariate time series. For the Earthquakes dataset, it is about the location and time of all earthquakes in Japan from 1990 to 2020 with magnitude of at least 2.5 from U.S. Geological Survey (2020). We follow the same preprocessing procedure for this dataset as Li (2023).

| Method | Training Time | Inference Time |
|---|---|---|
| Latent Diffusion | 2hr 10min | 5min 12s |
| Our Framework | 2hr 50min | 5min 17s |

Table 4: Comparison of Training and Inference Times on the MIMIC dataset.

| Method | Backbone | Retail | Energy |
|---|---|---|---|
| Latent Diffusion | Transformer | 0.037 | 0.052 |
| Latent Diffusion w/ Skip Connections | Transformer | 0.033 | 0.043 |
| Our Framework | Transformer | **0.025** | **0.031** |
| Latent Diffusion | LSTM | 0.041 | 0.057 |
| Latent Diffusion w/ Skip Connections | LSTM | 0.035 | 0.047 |
| Our Framework | LSTM | **0.027** | **0.033** |

Table 5: Comparison on two new time-series datasets, with another metric: MMD.

We use almost the same model configurations for all experiments. The diffusion models are parameterized by a standard U-net (Ronneberger et al., 2015), with $L = 1000$ diffusion iterations and hidden dimensions $\{128, 64, 32\}$. The hidden dimensions of autoencoders and latent variables are fixed as 128. The conditional distribution $p^{\text{gen}}(\mathbf{X} \mid \mathbf{z})$ is parameterized as a Gaussian, with learnable mean vector and diagonal covariance matrix functions. For our framework, $N, M$ are respectively selected as $50, 100$, with $\gamma = 2$ and $\eta = 1$. We also apply dropout with a ratio of $0.1$ to most layers of neural networks. We adopt Adam algorithm (Kingma & Ba, 2015) with the default hyperparameter setting to optimize our model. For Table 1 and Table 3, every number is averaged over 10 different random seeds, with a standard deviation less than $0.05$. For the computing resources, all our models can be trained on 1 GPU (40G memory) within 10 hours. Finally, the performance gains achieved by our models are all verified by the Student's t-test (Mishra et al., 2019).

# F   ADDITIONAL EXPERIMENTS

Due to the limited space of our main text, we put the results of some minor experiments here in the appendix. Notably, we will adopt more datasets and another evaluation metric.

## F.1   ABLATION STUDIES

We have conducted ablation studies to verify that our hyper-parameter selections $N = 50, M = 100$ are optimal. The experiment results are shown in Table 3. For both $N$ and $M$, either increasing or decreasing their values results in worse performance on the two datasets.

## F.2   STUDY ON RUNNING TIMES

Our framework only incurs a minor increase in training time and enjoys the same inference speed as the latent diffusion. For training, while our framework will run the decoder a second time for collapse simulation $\mathcal{L}^{\text{CS}}$, it can be made in parallel with the first run of decoder $f^{\text{dec}}$ for likelihood computation $\mathcal{L}^{\text{VI}}$. Therefore, the training is still efficient on GPU devices. Our framework also has a different way of variational inference to infer latent variable $\mathbf{z}$ from data $\mathbf{X}$. However, it admits a closed-form solution and is thus as efficient as the reparameterization trick of latent diffusion. For inference, our framework has no difference from the latent diffusion: sampling the latent variable $\mathbf{z}$ with the reverse diffusion process and running the decoder $f^{\text{dec}}$ in one shot.

To show the running times in practice, we perform an experiment on the MIMIC dataset as shown in Table 4. We can see that our framework indeed only has a minor increase for training. Given that our framework is free from posterior collapse and delivers better generation performances, this slight time investment is well worth it.

| Method | Backbone | ATIS | SNIPS |
|---|---|---|---|
| Latent Diffusion | Transformer | 37.12 | 59.36 |
| Latent Diffusion w/ Skip Connections | Transformer | 40.56 | 65.41 |
| Our Framework | Transformer | **51.73** | **78.12** |
| Latent Diffusion | LSTM | 35.38 | 55.72 |
| Latent Diffusion w/ Skip Connections | LSTM | 39.27 | 60.31 |
| Our Framework | LSTM | **48.46** | **71.45** |

Table 6: Performance comparison on two text datasets, with BLEU as the metric.

| Method | Backbone | CIFAR-10 |
|---|---|---|
| Latent Diffusion | U-Net | 3.91 |
| Latent Diffusion w/ KL Annealing | U-Net | 3.87 |
| Our Framework | U-Net | **3.85** |

Table 7: Performance comparison on an image dataset, with FID as the metric.

### F.3 MORE DATASETS AND ANOTHER EVALUATION METRIC

We conduct additional experiments on 2 more public UCI time-series datasets (Bay et al., 2000): Retail and Energy, with another widely used evaluation metric: maximum mean discrepancy (MMD) (Dziugaite et al., 2015). Lower MMD scores indicate better generative models. From the results shown in Table 5. We can see that our framework still significantly outperforms the baselines in terms of all the new benchmarks. For example, with LSTM as the backbone, our framework achieves a score that is $29.79\%$ lower than Skip Connections on the Energy dataset.

### F.4 MORE DATA MODALITIES

While our paper primarily focused on time series data, our framework is generally applicable to other types of data, including your mentioned text and images.

**Experiment on text data.** For text data, considering that natural language sentences exhibits a sequential structure similar to time series, it is intuitive that the posterior of text latent diffusion might also collapse. This intuition is supported by many evidences from previous works (Bowman et al., 2016; Fu et al., 2019). To verify that our framework is capable of improving text latent diffusion, we have conducted an experiment using two publicly available text datasets: ATIS (Hemphill et al., 1990) and SNIPS (Coucke et al., 2018).

The numbers in this table represent BLEU scores (Papineni et al., 2002), a widely used metric for evaluating text generation models. Higher scores indicate better performance. As the results shown in Table 6, we can see that our framework has significantly improved the text latent diffusion and notably outperformed a strong baseline—Skip Connections—across all datasets and backbones. Therefore, our framework also applies to text data.

**Experiment on image data.** For image data, we provide a detailed discussion in Sec. 4.1 of our paper: Image latent diffusion is rarely affected by posterior collapse because of its non-autoregressive decoder. To confirm this claim in practice, we have conducted an experiment comparing latent diffusion with our framework on the widely used CIFAR-10 dataset (Krizhevsky et al., 2009).

The results are shown in Table 7. The numbers in this table represent FID scores (Naeem et al., 2020), a common metric for evaluating image generation models. Lower scores indicate better performance. Our results show that both the baseline model (i.e., KL Annealing) and our framework improve the image latent diffusion to some extent. However, the improvements are minor, suggesting that image models are almost free from posterior collapse.

### F.5 CASE STUDIES OF DEPENDENCY MEASURES

Fig. 3 shows several examples of applying the dependency measures, where each subfigure contains a sample of time series (i.e., blue curve) generated by some model and two types of dependency

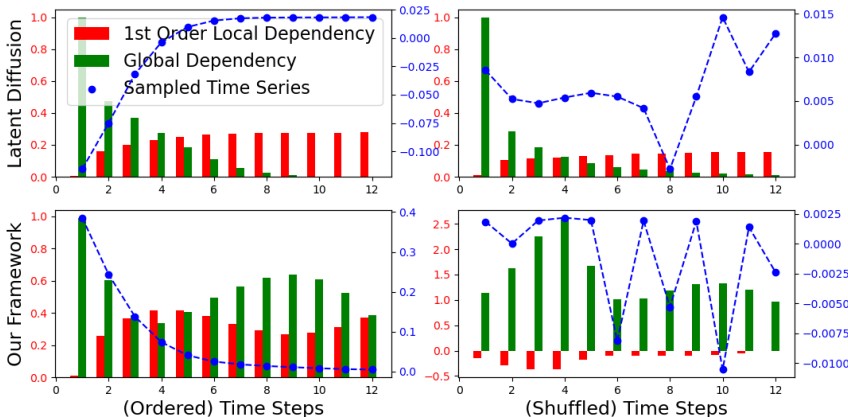

Figure 3: The global and local dependency measures $m_{t,0}, m_{t,t-1}$ (as defined in Sec. 3.2) respectively quantify the impacts of latent variable $\mathbf{z}$ and observation $\mathbf{x}_{t-1}$ on predicting the next one $\mathbf{x}_t$. We can see that the latent variable $\mathbf{z}$ of latent diffusion loses control over the condition generation $p^{\text{gen}}(\mathbf{X} \mid \mathbf{z})$, with *dependency illusion* (as introduced in Sec. 3.3) in the case of shuffled time series. In contrast, our framework has no such symptoms of *posterior collapse*.

measures (i.e., red and green bar charts) estimated by Eq. (9). Specifically, every point $\mathbf{x}_t$ in the time series corresponds to a green bar that indicates the global dependency $m_{t,0}$ and a red bar that represents the first-order local dependency $m_{t,t-1}$. In the upper left subfigure, we can see that the positive impact of latent variable $\mathbf{z}$ on the decoder (e.g., $m_{t,0}$) decreases over time and vanishes eventually. From the lower right subfigure, we can even see that some bars (i.e., local dependency $m_{t,t-1}$) are negative, indicating that the variable $\mathbf{x}_{t-1}$ provides the decoder with rather false information in predicting the next observation $\mathbf{x}_t$.

An example (i.e., the green bar chart) is shown in the upper left subfigure of Fig. 3. More interestingly, the upper right subfigure illustrates a phenomenon we call *dependency illusion*: Even when the time series is randomly shuffled and thus lacks structural dependencies, the decoder of latent diffusion still heavily relies on input observations (instead of the latent variable) for prediction (i.e., the global dependency measures $m_{t,0} = 1 - \sum_{s<t} m_{t,s}$ converges to 0 as time step $t$ increases). As demonstrated in the lower two subfigures of Fig. 3, our framework exhibits no signs of posterior collapse, such as the vanishing impact of latent variables over time. Note that the dataset that is used to train the baseline and our model is constructed by extracting the observations of the first 12 hours, with the top 1 feature that has the highest variance to form univariate time series.

