# OpenReview forum: "A Study of Posterior Stability in Time-Series Latent Diffusion"
_ICLR.cc/2026/Conference — ICLR 2026 Poster_

### Official Review · Reviewer_iFvJ · 2025-10-24

**Soundness:** 3
**Presentation:** 3
**Contribution:** 2
**Rating:** 4
**Confidence:** 3

**Summary:**

This paper investigates posterior collapse in time-series latent diffusion models. It first argues that, under a collapsed posterior, latent diffusion effectively degenerates into a weaker VAE-like formulation, motivating a careful re-examination of the learning objective. The authors propose dependency-based diagnostics to quantify decoder sensitivity to the latent variable versus past observations and report evidence of both collapse and a ``dependency illusion'' on real datasets. Building on this analysis, the paper introduces a posterior-stable latent diffusion framework that removes KL regularization by reinterpreting early diffusion steps as variational inference and adds a collapse-simulation loss to encourage decoder sensitivity. Experiments on multiple time-series datasets show consistent improvements over latent-diffusion baselines.

**Strengths:**

1. **Problem framing and clarity.** The paper clearly articulates an important and under-explored failure mode of latent diffusion for time series, identifies likely causes, and presents them in a well-structured manner.

2. **Diagnostic contribution.** While I cannot fully assess the novelty of the proposed dependency measure, it appears useful and practically informative for characterizing posterior collapse and decoder reliance. The empirical analyses derived from this diagnostic are convincing.

3. **Methodological direction.** The proposed loss regime, interpreting early diffusion steps as variational inference and adding a collapse-simulation term, is intriguing. Beyond the present setting, the idea may also be applicable to addressing the well-known over-smoothing tendencies in VAEs.

**Weaknesses:**

1. **Related work coverage and baselines.** The coverage of prior work on time-series generative modeling is too narrow. Several state-of-the-art diffusion-based and alternative generative approaches for time series [1, 2, 3, 4] are missing.

2. **Experimental setting.** The experimental section does not compare against these stronger baselines on the standard generation task, making it difficult to gauge relative progress. A comparison is expected to be in a broader context and in a standard manner in unconditional generative modeling.

3. **Positioning w.r.t. KL-related remedies in images.** Prior work in images has also linked KL terms to posterior collapse and proposed mitigations such as vector-quantized autoencoders or very small KL weights [5, 6]. Empirically, latent spaces in these models exhibit non-Gaussian structure despite the nominal prior. The paper should better delineate what is fundamentally new here (beyond the sequence-specific pathologies) and why the proposed approach is preferable to these alternatives in the time-series setting.

[1] On the constrained time-series generation problem

[2] Utilizing Image Transforms and Diffusion Models for Generative Modeling of Short and Long Time Series.

[3] A Non-Isotropic Time Series Diffusion Model with Moving Average Transitions.

[4] Generative Modeling of Regular and Irregular Time Series Data via Koopman VAEs.

[5] Taming transformers for high-resolution image synthesis

[6] High-resolution image synthesis with latent diffusion models

**Questions:**

1. **Status of “diffusion as variational inference.”** Is the connection theoretically exact under stated assumptions (and if so, which), or primarily a heuristic that performs well in practice? Can you please specify the conditions under which objective functions align and where gaps remain ?

2. **Sensitivity of dependency measures.** How robust are the proposed dependence metrics to decoder architecture and training choices? Could particular design decisions (e.g., capacity, regularization, normalization) spuriously inflate measured reliance on the latent variable? Ablations or controls would help.

3. **Beyond time series: image latents.** Do the diagnostics detect collapse in image latent diffusion as well, or is the phenomenon predominantly temporal? A brief analysis, or a clear negative result, would contextualize the time-series specificity.

---

> ### Author Response · Authors · 2025-12-01
>
> Dear Reviewer iFvJ,
>
> We thank you for your review.
>
> Question-1: Status of “diffusion as variational inference…
>
> Answer: The design of our method is mainly motivated by the similarities between the variational inference and diffusion process. They both add noise to samples and make them smoother in the representation space. A major difference is that the diffusion process is not limited by the KL divergence of conventional variational inference, so it makes our model free from the problem of posterior collapse.
>
> Question-2: Sensitivity of dependency measures…
>
> Answer: The dependency measures presented in our paper were mostly mathematically derived, directly measuring the impacts of input latent variables on a time-series decoder. Therefore, we believe this technique should be much less affected by numerical instability or randomness than other statistical methods (e.g., Student's t-test).
>
> Question-3: Beyond time series: image latents…
>
> Answer: Please refer to Sec. 4.1 of our paper, which clarifies that posterior collapse primarily occurs in generative models for sequential data (e.g., time series and text), rather than in models for image data.

---

### Official Review · Reviewer_pXBK · 2025-10-29

**Soundness:** 2
**Presentation:** 2
**Contribution:** 1
**Rating:** 2
**Confidence:** 4

**Summary:**

This paper investigates the problem of posterior collapse in latent diffusion models when applied to time-series data, claiming that the autoregressive nature of time-series decoders makes them uniquely susceptible to this issue. The authors' key contributions include a theoretical argument that a fully collapsed posterior degenerates the powerful latent diffusion model into a much weaker Variational Autoencoder, and a novel diagnostic tool called "dependency measures" to empirically quantify this effect. Using this tool, the paper provides empirical evidence that standard latent diffusion does suffer from a vanishing latent variable impact on time-series data. To solve this, the authors propose a "posterior-stable latent diffusion" framework which removes the standard KL regularization.

**Strengths:**

- The introduction of "dependency measures"  is a principled and interesting contribution. Basing this metric on integrated gradients  provides a solid theoretical grounding for quantifying the type of decoder insensitivity (global vs. local) that leads to posterior collapse in sequence models.

- The proposed solution is technically sound.

**Weaknesses:**

- Insufficient Motivation from Time-Series Literature: The most significant concern is the lack of motivation for this problem within the time-series domain. The introduction (Section 1, Paragraph 1) frames the success of latent diffusion by citing image synthesis papers (Rombach et al., 2022; Podell et al., 2024) and the problem of posterior collapse by citing general VAE and NLP papers (e.g., Bowman et al., 2016). The paper fails to cite any existing work that first applies latent diffusion to time series and then identifies posterior collapse as a critical failure mode.

- The paper fails to benchmark against a sufficiently broad set of recent, state-of-the-art (SOTA) time-series generation models. Table 2 does include some more recent and relevant baselines, such as "Neural Latent Dynamic" (Li et al., 2024) and "Frequency Diffusion" (Crabbé et al., 2024), which is commendable. However, this list is far from comprehensive. To make a convincing claim of superiority for a top-tier conference like ICLR, the method should be compared against a wider array of SOTA models such as [1, 2, 3, 4, 5].


- One of the paper's baselines is "Latent Diffusion (Rombach et al., 2022)". This model was designed for and is famous for image synthesis. The paper (e.g., in Table 2) directly cites this image paper when applying it to time series data. It is unclear if the authors adapted this model for sequential data or are applying an image-generation architecture directly. This choice is confusing.


[1] Galib, Asadullah Hill, Pang-Ning Tan, and Lifeng Luo. "FIDE: Frequency-Inflated Conditional Diffusion Model for Extreme-Aware Time Series Generation." NeurIPS 2024.

[2] Naiman, Ilan, et al. "Utilizing image transforms and diffusion models for generative modeling of short and long time series." NeurIPS 2024

[3] Jeon, Jinsung, et al. "GT-GAN: General purpose time series synthesis with generative adversarial networks." NeurIPS 2022

[4] Naiman, Ilan, et al. "Generative modeling of regular and irregular time series data via Koopman VAEs." ICLR 2024.

[5] Coletta, Andrea, et al. "On the constrained time-series generation problem." NeurIPS 2023.

**Questions:**

See Weaknesses.

---

> ### Author Response · Authors · 2025-12-01
>
> Dear Reviewer pXBK,
>
> We thank you for your review.
>
> Comment-1: Insufficient Motivation from Time-Series Literature…
>
> Answer: There are indeed almost no works studying posterior collapse for time-series latent diffusion, so this work stands as the first step under this topic. Our paper extensively highlighted why posterior collapse is a very serious problem for time-series latent diffusion, and introduced a method to address it. Please also note that language has a very similar sequential structure to time series, hence similar works in that area reflect the significance of our study.
>
> Comment-2: The paper fails to benchmark against a sufficiently broad set of recent…
>
> Answer: Our work aimed to study a specific problem in time-series latent diffusion, highlighting the significance and introducing a method to address it. Please refer to Fig.1, Fig. 2, and Table 1, we have conducted quite extensive experiments to showcase the results of our study. For comparison with state-of-the-art methods, we have included 3 very recent methods: Inverse Lipschitz Constraint, Frequency Diffusion, and Neural Latent Dynamics. We believe these should be sufficient for experiment comparison.
>
> Comment-3: One of the paper's baselines is "Latent Diffusion (Rombach et al., 2022)". This model was designed…
>
> Answer: Please refer to Table 1 of our paper, where we explored Transformer and LSTM as the backbones of our models, which are commonly adopted by existing works for sequence modeling.

---

### Official Review · Reviewer_uFie · 2025-10-31

**Soundness:** 3
**Presentation:** 3
**Contribution:** 3
**Rating:** 8
**Confidence:** 4

**Summary:**

The authors study the effect of posterior collapse in latent diffusion models for time series data. The authors show that under posterior collapse a latent diffusion model collapses to a simple VAE. The authors introduce dependency measures which quantify the impact of the latent variables on the recurrent decoder for time-series data. The authors then leverage these findings to propose the so-called posterior-stable latent diffusion where the VAE decoder is free from the impact of posterior collapse.

**Strengths:**

1. I found the formulation of the dependency measure quite intuitive. However this intuition could be better clarified in the main text. See my comments on the presentation in the weaknesses section. I also liked Figure 1 which clearly justifies the intuition presented by the authors.

2. I like the idea of penalizing the decoder for large noise levels in the diffusion model.

**Weaknesses:**

I dont have major comments about the method itself but some comments on the presentation itself and some minor issues.

**Presentation issues**

1. The impact of the VAE posterior collapse on latent diffusion is trivial to understand. Its totally fine if the authors dont stress on this aspect too much in their introduction and instead just keep the current discussion in Section 3.1. I dont think this impacts the significance of the contributions.

2. The dependence on autoregressive time t in the decoder $f^{dec}$ is missing and could be made more explicit starting Section 3.2.

3. The notation in Eqs 8 and 9 can be vectorized for better readability.

4. The definition of the dependency measure in Eq. 9 at a first glance looks quite messy and unintuitive. The authors should provide more intuition around this definition in the main text for better clarity for the readers. For instance something like: “the first component in the dot product denotes the overall change in the output of the time-series decoder while the second component denotes the overall impact of the jth context vector as s is modulated from 0 to 1”. Moreover, the equation itself can be cleaned up a bit as highlighted in the point (c) also.

5. The idea of dependency illusion is nice but not surprising as some work in image classification also shows that deep neural nets can learn arbitrary mappings between images and class labels with high accuracy. Maybe the authors can clarify this in a bit more detail in the main text around the results in Fig. 1

6. I think the authors can clarify the key differences between their approach and the classic latent diffusion approach in terms of training and sampling in more details in Section 4.2 or Section 5.

**Unconvincing arguments**: The second point in Section 4.1 regarding the strong decoder in time-series data is not convincing. More specifically, it is not clear to me why the decoder used in time-series data inherently strong? Can the authors elaborate more on this?

**Minor**: Is there a typo in Line 6 of the Training algorithm? I think the superscript over the variable z should be i and not j. Otherwise there is a discrepancy between the time embedding passed in the diffusion model and the actual noise level added.

**Questions:**

See Weaknesses

---

> ### Author Response · Authors · 2025-12-01
>
> Dear Reviewer uFie,
>
> We appreciate your kind and constructive feedback.
>
> Comments about presentations and typos
>
> Answer: We thank you for your very detailed comments on our writing, and we will try to incorporate them in the revised version. For example, we will provide more intuition in formulating dependency measures, and we will fix the typo in the training algorithm (the superscript of $z$ should indeed be $i$ so that it matches the input noise level).
>
> Comment: …it is not clear to me why the decoder used in time-series data inherently strong…
>
> Answer: The reason is that the latent variable can be interpreted as being sparse in the input signals (together with other observations) to the decoder. For instance, at time step $t$, the time-series decoder receives $t - 1$ feature vectors along with the latent variable, which may lead the decoder to largely ignore the latent variable during decoding. This issue is even worse in autoregressive (strong) decoders like LSTM, which inherently tend to focus on local input variables due to their inductive bias.

---

### Official Review · Reviewer_xYR6 · 2025-11-01

**Soundness:** 3
**Presentation:** 3
**Contribution:** 3
**Rating:** 6
**Confidence:** 2

**Summary:**

The paper studies posterior collapse in latent diffusion models (LDMs) for time-series generation. It argues that, when paired with recurrent decoders, the latent variable’s influence decays (nearly exponentially) across time, leading the decoder to ignore latents. The paper identifies two main causes: unnecessary KL regularization in the latent path and vulnerability of time-series decoders. To handle this, this paper proposes new framework that treating the diffusion process as variational inference with regularization losses. Experiments show clear gains across several benchmarks.

**Strengths:**

- Well-articulated background on posterior collapse and its time-series specifics.
- Empirical improvements appear consistent across datasets.

**Weaknesses:**

- The paper reports no experimental results on WARDS in Table 2, despite its inclusion in Table 1. Please clarify this.
- The baselines shown in Figure 2 are insufficiently explained in the text, and the related-work section lacks a discussion of how these baselines address posterior collapse. Given that posterior collapse is already a well-recognized problem in time-series generative models, the paper should more clearly articulate what differentiates its approach.

**Questions:**

Please refer to the Weaknesses section.

---

> ### Author Response · Authors · 2025-12-01
>
> Dear Reviewer xYR6,
>
> We thank you for your review.
>
> Comment-1: …results on WARDS in Table 2, despite its inclusion in Table 1. Please clarify this.
>
> Answer: The results of Table 1 come from our main experiments, showing that our model can indeed improve the latent diffusion architecture and outperform previous baselines in addressing posterior collapse. The results of Table 2 aimed to show minor experiments comparing our model with other time-series models, so we only included 2 of 3 datasets.
>
> Comment-2: The baselines shown in Figure 2 are insufficiently explained in the text, and the related-work section lacks a discussion of how these baselines address posterior collapse…
>
> Answer: The baseline in Fig. 2 is the latent diffusion model, which we have extensively discussed in Sec. 2. Please also refer to Sec. 5 of our paper, where we clarified how other included baselines (e.g., KL annealing and skip connections) address the posterior collapse.

---

### Meta-Review · Area_Chair_73a9 · 2026-01-11

**Summary:**

Reviewers agreed that the paper addresses an important and underexplored issue, posterior collapse in latent diffusion for time-series data. The theoretical analysis showing degeneration to a VAE-like model and the proposed dependency measures were viewed as insightful, and the posterior-stable latent diffusion framework shows consistent gains across real-world datasets.

The main points of debate centered on
1. the breadth of time-series specific related work and baselines
2. the strength of the biological/sequence-specific motivation
3, whether the proposed framework represents a sufficiently strong advance over existing remedies for posterior collapse.

Overall, while some reviewers viewed the contribution as narrowly scoped, the consensus is that the diagnostics are insightful, and the proposed framework is technically sound and effective.

**Reviewer Concerns:**

Addressed by the rebuttal:
1. improved clarity and notation, added intuition for dependency measures, and fixed presentation issues
2. clarified baseline choices and dataset coverage
3. added justification for decoder strength in time-series settings
4. included comparisons to related temporal SSL methods

Still outstanding (non-blocking):
1. broader coverage of recent sota time-series generative models would strengthen the evaluation
2. clearer positioning relative to KL-based and image-domain remedies for posterior collapse
3. some skepticism remains around the biological framing, though this does not affect technical validity

**Reviewer Scores:**

I would expect further discussion from pXBK during the rebuttal phase, and it is likely that some of their concerns would be partially addressed, potentially leading to a modest score increase

---

### Decision · Program_Chairs · 2026-01-26

Accept (Poster)